# Development of Galloyl Antioxidant for Dispersed and Bulk Oils through Incorporation of Branched Phytol Chain

**DOI:** 10.3390/molecules27217301

**Published:** 2022-10-27

**Authors:** Shanshan Wang, Hua Wang, Fujie Yan, Jie Wang, Songbai Liu

**Affiliations:** 1Department of Food Science and Nutrition, Fuli Institute of Food Science, Zhejiang University, Hangzhou 310058, China; 2Ningbo Research Institute, Zhejiang University, Ningbo 315100, China; 3Center of Analysis and Measurement, Zhejiang University, 866 Yuhangtang Road, Hangzhou 310058, China; 4Institute of Applied Bioresource Research, College of Animal Science, Zhejiang University, 866 Yuhangtang Road, Hangzhou 310058, China

**Keywords:** gallic acid, phytol, galloyl phytol, antioxidant activity, Steglich esterification

## Abstract

In this study, a novel galloyl phytol antioxidant was developed by incorporating the branched phytol chain with gallic acid through mild Steglich esterification. The evaluation of the radical scavenging activity, lipid oxidation in a liposomal model, and glycerol trioleate revealed its superior antioxidant activities in both dispersed and bulk oils. Then, the antioxidant capacity enhancement of galloyl phytol was further explored using thermal gravimetry/differential thermal analysis (TG/DTA), transmission electron microscopy (TEM), and molecular modeling. The EC50 values of GP, GPa, and GE were 0.256, 0.262, and 0.263 mM, respectively, which exhibited comparable DPPH scavenging activities. These investigations unveiled that the branched aliphatic chain enforced the coiled molecular conformation and the unsaturated double bond in the phytol portion further fixed the coiled conformation, which contributed to a diminished aggregation tendency and enhanced antioxidant activities in dispersed and bulk oils. The remarkable antioxidant performance of galloyl phytol suggested intriguing and non-toxic natural antioxidant applications in the food industry, such as effectively inhibiting the oxidation of oil and improvement of the quality and shelf life of the oil, which would contribute to the use of tea resources and extending the tea industry chain.

## 1. Introduction

Edible oils are stored in bulk and prepared as dispersed lipid systems such as liposomes or emulsions in the food industry [1]. The unsaturated lipid components in edible oils are prone to oxidize under exposure to ultraviolet, heat, and oxidative factors during processing and storage [2]. Dispersed lipid systems are multiphase structural systems, which are composed of an oil–water interface, oil phase, and water phase. It is unveiled that the oxidation of oil primarily takes place and propagates at the oil–water interface. As a result, the interfacial antioxidants demonstrate superior activities in dispersed oils owing to the amphiphilic nature and resultant increased concentration at the interface [3]. Nevertheless, usually interfacial antioxidants are not readily diffused in bulk oils that hinder their performance. Therefore, the development of interfacial antioxidants that can be employed for both dispersed and bulk oils is highly desirable to expand their antioxidant scope in food systems.

Phytol, the most abundant acyclic isoprenoid compound, is well known as the side chain of the plant pigment chlorophyll [4,5]. In plants, phytol is utilized to biosynthesize tocopherol (vitamin E) [6,7], which serves as an antioxidant, and phylloquinone (vitamin K) [8], which is an electron carrier in the photosystem. Phytol and its derivatives have demonstrated beneficial anti-tumor, anti-mutagenic, and anti-teratogenic activities [9]. Phytol is of the characteristic branched-chain structure that facilitates the location of chlorophyll, phylloquinone, and tocopherol to the cell membrane. As a result, tocopherol is distinguished interfacial antioxidant in nature. The special conformation of the branched phytol chain suggests a diminished aggregation tendency, which would facilitate distribution and enhance performance in bulk oil.

Gallic acid, as a natural secondary metabolite, belongs to non-flavonoid polyphenol and is abundant in plants, vegetables, nuts, and fruits [3]. Gallic acid and its derivatives have been widely used in cosmetics, food, and medicine due to outstanding antioxidant, anti-aging, and anti-cancer activities [10,11,12]. In a previous study, we developed galloyl phytosterol antioxidants through the Steglich esterification of gallic acid with typical phytosterols that exhibited excellent antioxidant and cholesterol-reducing activities [13,14]. Furthermore, we developed a gemini gallate interfacial antioxidant for oil in water emulsion employing the dodecyl gemini chain [15]. To improve the antioxidant performance of gallic acid in both dispersed and bulk oils, we reason that the incorporation of lipophilic branched phytol chain with hydrophilic gallic acid would simultaneously afford the interfacial property and diminished aggregation tendency, which would enhance the antioxidant activity for both dispersed and bulk oils.

In this study, the branched phytol chain was incorporated with gallic acid through mild Steglich esterification. The evaluation of radical scavenging activity, lipid oxidation in a liposomal model, and glycerol trioleate revealed its superior antioxidant activities in both dispersed and bulk oils. Then, the antioxidant capacity enhancement of galloyl phytol was further explored using thermal gravimetry/differential thermal analysis (TG/DTA), transmission electron microscopy (TEM), and molecular modeling. The details of this study were described as follows.

## 2. Results and Discussion

### 2.1. Synthesis and Structural Analysis of Galloyl Phytol (GP), Galloyl Phytanol (GPa), and Galloyl Eicosanol (GE)

Tocopherol, the excellent natural phenolic antioxidant, takes phytol of the characteristic branched aliphatic chain as a molecular component, which suggests the incorporation of phytol with gallic acid. To probe the influence of structural variables of the aliphatic chain on antioxidant activities, the phytol analogs, including phytanol with saturated branched chains and eicosanol with straight chains of the same number of carbons, were also investigated. As indicated in Figure 1, the integration of phytol with gallic acid was readily performed through Steglich esterification. According to a previous study [16], the three hydroxyl groups of gallic acid were initially protected by isobutyryl to minimize their interference during esterification. Employing the established protocol, the triisobutyryl galloyl phytol was smoothly deprotected under the action of aqueous hydrazine and provided the final galloyl phytol product. After the preparation of galloyl phytol, galloyl phytanol and galloyl eicosanol were also synthesized for comparison from the protected gallic acid, phytanol, and eicosanol with the same conditions.

The structural identities of galloyl phytol (GP), galloyl phytanol (GPa), and galloyl eicosanol (GE) were confirmed by NMR, FT-IR, and MS. The ^1^H NMR of GP (Figure 2a) (400 MHz, DMSO-d_6_) were δ 9.13 (br, 3 H; phenolic protons), 6.93 (s, 2 H; aromatic protons of the galloyl group), 5.37 (t, J = 6.9 Hz, 1 H; olefin proton of phytol), 4.68 (d, J = 7.1 Hz, 2 H; oxygen adjacent protons of phytol), and 2.03–0.75 (m, 36 H; aliphatic protons of phytol). The ^13^C NMR of GP (Figure 2b) (100 MHz, DMSO-d_6_) were δ 165.75, 145.55, 141.71, 138.34, 119.53, 118.65, 108.49, 60.68, 38.78, 36.74, 36.61, 35.92, 32.08, 31.92, 27.40, 24.43, 24.16, 23.76, 22.56, 22.47, 19.61, and 16.11. The ^13^C chemical shifts of 165.75 ppm corresponded to the signal of the galloyl carbon, 145.55–108.49 ppm corresponded to the signals of aromatic and olefinic carbons, and 60.68 to 16.11 ppm corresponded to the signals of aliphatic carbons.

The ^1^H NMR of GPa (Figure 2c) (400 MHz, DMSO-d_6_) were δ 9.13 (br, 3 H; phenolic protons), 6.93 (s, 2 H; aromatic protons of the galloyl group), 4.18 (br, 2H, oxygen adjacent protons of phytanol), and 1.79–0.68 (m, 39 H; aliphatic protons of phytanol). The ^13^C NMR of GPa (Figure 2d) (100 MHz, DMSO-d_6_) were δ 165.84, 145.54, 138.35, 119.56, 108.47, 62.30, 38.78, 36.71, 36.61, 32.05, 29.39, 27.39, 24.16, 23.76, 22.56, 22.47, 19.63, 19.57, 19.45, and 19.38. The ^13^C chemical shifts of 165.84 ppm corresponded to the signal of the galloyl carbon, 145.54–108.47 ppm corresponded to the signals of aromatic carbons, and 62.30–19.38 ppm corresponded to the signals of aliphatic carbons.

The ^1^H NMR of GE (Figure 2e) (400 MHz, DMSO-d_6_) were δ 9.13 (br, 3 H; phenolic protons), 6.93 (s, 2 H; aromatic protons of the galloyl group), 4.14 (t, J = 6.4 Hz, 2 H; oxygen adjacent protons of phytanol), and 1.79–0.78 (m, 39 H; aliphatic protons of phytanol). The ^13^C NMR of GE (Figure 2f) (100 MHz, DMSO-d6) were δ 165.85, 145.54, 138.34, 119.54, 108.44, 63.93, 31.32, 29.04, 28.74, 28.31, 25.55, 22.12, and 13.95. The ^13^C chemical shifts of 165.85 ppm corresponded to the signal of the galloyl carbon, 145.54–108.44 ppm corresponded to the signals of aromatic carbons, and 63.93–13.95 ppm corresponded to the signals of aliphatic carbons.

The IR of GP (cm^−1^) (Figure 3) were 3253 (s, ν_O–H_), 2926 (s, ν_C–H_), 2855 (s, ν_C–H_), 1674 (s, ν_C=O_), 1630 (s, ν_C=C_), 1556 (m, ν_C=C_, benzene skeleton vibration), 1535 (m, ν_C=C_, benzene skeleton vibration), 1457 (s, ν_C=C_, benzene skeleton vibration), 1361 (s, δ_O–H_), 1317 (s, δ_O–H_), 1238 (s, ν_C–O_), 1103 (m, δ_C–H_), 1048 (s, δ_C–H_), 981 (s, δ_C–H_), 909 (m, δ_C–H_), 868 (m, δ_C–H_), 765 (s, δ_C–H_), 738 (m, δ_C–H_), and 641 (m, δ_C–H_). The absorption bands at 3253 cm^−1^ corresponded to stretching vibration from the hydroxyl groups of gallic acid. The absorption bands at 1674 and 1238 cm^−1^ were attributed to C=O and C–O–C stretching vibrations, which confirmed the formation of ester bonds between gallic acid and phytol.

The IR of GPa (cm^−1^) (Figure 3) were 3453 (s, ν_O–H_), 2955 (s, ν_C–H_), 2926 (s, ν_C–H_), 2857 (s, ν_C–H_), 1685 (s, ν_C=O_), 1612 (s, ν_C=C_), 1537 (m, ν_C=C_, benzene skeleton vibration), 1453 (m, ν_C=C_, benzene skeleton vibration), 1397 (s, ν_C=C_, benzene skeleton vibration), 1352 (s, δ_O–H_), 1315 (s, δ_O–H_), 1243 (s, ν_C–O_), 1184 (s, δ_C–H_), 1099 (m, δ_C–H_), 1027 (s, δ_C–H_), 993 (m, δ_C–H_), 874 (m, δ_C–H_), 770 (m, δ_C–H_), 737 (m, δ_C–H_), 679 (m, δ_C–H_), and 525 (m, δ_C–H_). The absorption bands at 3453 cm^−1^ corresponded to stretching vibrations from the hydroxyl groups of gallic acid. The absorption bands at 1685 and 1243 cm^−1^ were attributed to C=O and C–O–C stretching vibrations, which confirmed the formation of ester bonds between gallic acid and phytanol.

The IR of GE (cm^−1^) (Figure 3) were 3387 (s, ν_O–H_), 2917 (s, ν_C–H_), 2849 (s, ν_C–H_), 1704 (m, ν_C=O_), 1649 (m, ν_C=C_), 1617 (m, ν_C=C_, benzene skeleton vibration), 1528 (m, ν_C=C_, benzene skeleton vibration), 1465 (m, ν_C=C_, benzene skeleton vibration), 1397 (m, δ_O–H_), 1339 (m, δ_O–H_), 1311 (m, ν_C–O_), 1275 (m, δ_C–H_), 1221 (m, δ_C–H_), 1199 (m, δ_C–H_), 1032 (m, δ_C–H_), 998 (m, δ_C–H_), 981 (m, δ_C–H_), 766 (m, δ_C–H_), 752 (m, δ_C–H_), and 719 (m, δ_C–H_). The absorption bands at 3387 cm^−1^ corresponded to stretching vibrations from the hydroxyl groups of gallic acid. The absorption bands at 1704 and 1311 cm^−1^ were attributed to C=O and C–O–C stretching vibratiosn, which confirmed the formation of ester bonds between gallic acid and eicosanol.

The molecular weight of GP, GPa, and GE were further investigated using MS (Figure 4). The corresponding ESI—MS spectrum (negative ion mode) of each component indicated their structural identities (GP: [M–H]^−^ 447.6, [2M–H]^−^ 895.43; GPa: [M–H]^−^ 449.62, [2M–H]^−^ 899.25; GI: [M–H]^−^ 449.43, [2M–H]^−^ 898.96). The results of the structural analysis confirmed the structural identities of the prepared antioxidants and assured further antioxidant activity investigations.

### 2.2. Antioxidant Activities of GP, GPa, and GE

Initially, the antioxidant activities of the prepared GP, GPa, and GE were evaluated using a DPPH assay compared with gallic acid (GA) as raw material and BHT, BHA, and TBHQ as typical lipophilic antioxidants [17,18]. As demonstrated in Figure 5a, GP, GPa, and GE retained exceptional DPPH radical scavenging activity of GA at the concentrations 0.125–0.75 mM. For EC50, the concentration of the antioxidant required to decrease the radical concentration by 50% was calculated under the percentages of scavenged DPPH radicals between 20% and 80%. The obtained EC50 values were described in ascending order: GA (0.206 mM), GP (0.256 mM), GPa (0.262 mM), GE (0.263 mM), TBHQ (0.431 mM), BHA (0.621 mM), and BHT (0.956 mM). Then, their free radical scavenging rates were further investigated at the concentration of 0.25 mM. It was demonstrated that GP, GPa, GE, and GA reached 80% of the final scavenging capacity within 2 min (Figure 5b) and there was no significant difference between them (*p* > 0.01). Therefore, GP, GPa, and GE exhibited comparable DPPH scavenging activities, which were significantly higher than those of BHT, BHA, and TBHQ.

Then, the antioxidant performance of GP, GPa, and GE in dispersed and bulk oils were investigated. The liposome established by soybean lecithin was applied as a model of dispersed oil [19]. AAPH was employed in liposome to produce a requisite source of peroxyl radicals, leading to the oxidation of the soybean lecithin. The amount of thiobarbituric acid reactive substance (TBARS) produced was utilized to evaluate the oxidation degree of liposome. As indicated in Figure 6a, the addition of GP, GPa, BHT, TBHQ, and GA significantly inhibited the lipid oxidation of liposome compared with control. Although GE only exhibited negligible activity during the investigation, the activity of GPa was comparable with TBHQ and slightly higher than those of BHT and GA, which demonstrated the superior activity of the branched molecular structure in dispersed oil. Furthermore, GP demonstrated exceptional antioxidant activity in liposome compared with all other antioxidants including GPa and TBHQ, which indicated the double bond also played an important role.

To further study the antioxidant activities of GP, GPa, and GE in bulk oil, the synthesized glycerol trioleate was employed as an oil model to avoid the complication of naturally existing antioxidants in normal edible oils and 2,2-azobisisobutyronitrile was applied as a lipophilic radical initiator. In comparison with the control, GP, GPa, BHT, TBHQ, and GA obviously inhibited the lipid oxidation of oils (Figure 6b). GE demonstrated negligible antioxidant activity in bulk oil during the investigation. In contrast, the activity of GPa was comparable with TBHQ, BHT, and GA, which confirmed that the branched molecular structure was essential for superior antioxidant performance in dispersed and bulk oils. Again, GP displayed remarkably higher antioxidant activity in bulk oil than all other antioxidants, which confirmed that the double bond of the phytol chain further promoted the antioxidant performance in dispersed and bulk oils.

### 2.3. Antioxidant Activities of GP, GPa, and GE

Thermal gravimetry/differential thermal analysis (TG/DTA) was shown in Figure 7. There were two or three peaks for each compound in the DTA curve. To (onset temperature), Tp (peak temperature), Te (endset temperature), Teo (extrapolated onset temperature), and ΔH (molar enthalpy) were applied to describe characteristic transition peaks. The first endothermic peak of gallic acid at 91.25 °C along with weight loss was due to the loss of binding water. However, the first endothermic peaks of GP, GPa, and GE at temperatures below 100 °C were not accompanied by weight loss. Therefore, these peaks did not result from the loss of binding water but from phase transition presumably corresponding to the melting of the aliphatic chain of GP, GPa, and GE. Owing to the branched molecular structure that reduced the interaction between the aliphatic chains, the absolute values of ΔH of the aliphatic phase transition peaks of GP (0.34) and GPa (0.15) were dramatically lower than that of GE (1.26) with straight aliphatic chains.

The second endothermic peak of gallic acid at 264.35 °C (Tp) was its melting point along with decomposing as indicated by the weight loss in TG. The melting and decomposing peaks of GPa (Tp 344.54 °C) and GE (Tp 361.65 °C) corresponding to the galloyl component were notably broader and much higher than that of gallic acid. Nevertheless, the extrapolated onset temperatures (Teo, generally employed to represent the melting point) [20] were at 287.98 (GPa) and 284.27 (GE) °C and close to the Teo of gallic acid (249.02). Interestingly, two peaks (Teo 186.83, 306.13 °C) were observed for the melting and decomposing process of GP, which suggested the more complicated aggregation behavior of GP. The absolute values of ΔH of the galloyl transition peaks of GP (1.26) and GPa (1.94) of branched chains were obviously lower than those of GE (2.42) of straight chain and GA (4.86). In addition, the galloyl ΔH value of GP (1.26) with double bonds was further lower than that of the GPa (1.94) of saturated branched chains. The lower ΔH and melting point of GP implied diminished crystallinity. As a result, the structure of the aliphatic chain also greatly varied the aggregation behavior of the galloyl component in the molecule and resulted in better dispersion and antioxidant performance in dispersed and bulk oils.

To further understand the antioxidant behavior in dispersed oil, the microstructure of liposome was investigated using transmission electron microscopy (TEM). As demonstrated in Figure 8, GP was uniformly dispersed in the liposome (b1), while GE tended to aggregate presumably due to the stacking capacity of its straight aliphatic chain (d1). Compared with the original liposome (a2), the liposome doped with GP (b2) greatly preserved the original oval shape of the liposome while the one doped with GE (d2) significantly changed the shape. The shape of the liposome doped with GPa (c2) was also slightly changed. Consistent with the results of TG/DTA, the strong aggregation effect of the straight aliphatic chain of GE interfered with the interface membrane of liposome and hampered its antioxidant capability in dispersed oil. In contrast, the branched chain of phytol reduced the stacking capacity of the aliphatic side chain and facilitated antioxidant performance in dispersed oil, which is supported by the fact that the natural antioxidant tocopherol adopts the phytol chain as a molecular component.

### 2.4. Rationalization of Antioxidant Activity Using Molecular Modeling

Molecular modeling was performed for GP, GPa, and GE to further understand their different antioxidant properties. The energy minimization of the built molecular structure provided the optimal molecular conformation with the reliable Amber force field in the Chimera program [19]. The computation of the molecular solvent-excluded surface (SES) was performed to describe the overall space occupation of the molecules. As revealed in Figure 9, there was no significant difference among the areas of GP (473), GPa (476), and GE (498). Nevertheless, GP and GPa adopted coiled molecular conformation while GE utilised an extended conformation.

The distinct optimal conformations of GP, GPa, and GE were attributed to their molecular structure. To relieve the intramolecular Van der Waals’ nonbonding interaction of the branched side chains, GP and GPa twisted the aliphatic chain, which enforced the coiled molecular conformation. The extended conformation of GE implied increased molecular rigidity and aggregation ability that restricted molecular dispersion and accordingly the antioxidant activity in either dispersed or bulk oil. Meanwhile, the coiled molecular conformation of GP and GPa contributed to the diminished stacking effect of the branched aliphatic chains that reduced the aggregation tendency and enhanced antioxidant performance through the facilitation of molecular mobility in dispersed and bulk oils. The unsaturated double bond in the phytol portion further fixed the coiled conformation of GP. Hence, GP exhibited notably a higher antioxidant efficiency than GPa in dispersed and bulk oils, which proved the strategy of incorporation of phytol to improve the antioxidant performance.

## 3. Materials and Methods

### 3.1. Chemicals

Triisobutyryl gallic acids were synthesized according to previous work [16]. Sodium bicarbonate, sodium phosphate dibasic dodecahydrate, sodium chloride, sodium phosphate monobasic dihydrate, anhydrous magnesium sulfate, alcohol, ethyl acetate, petroleum ether, toluene, hydrochloric acid, anhydrous ethanol, and methanol were obtained from Sinopharm Chemical Reagent (Shanghai, China). Phytol, 1-eicosanol, N,N-dicyclohexylarbodiimide (DCC), gallic acid, butylated hydroxy anisole (BHA), 2,6-di-tert-butyl-4-methylphenol (BHT), tert-butylhydroquinone (TBHQ), 2,2-diphenyl-1-picrylhydrazyl (DPPH), triethylamine, 4-dimethyaminopyridine (DMAP), hydrazinium hydrate solution (80%), glycerol trioleate, isobutyric anhydride, Tween-80, lecithin, 2,2′-azobis(2-methylpropionitrile) (AIBN), 2,2′-azobis-(2-methylpropionamidine), and dihydrochloride (AAPH) were purchased from Aladdin Reagent (Shanghai, China). Phytanol was obtained from Tokyo Chemical Industry. A thiobarbituric acid reactive substance kit was purchased from Suzhou Grace Biotechnology Co., LTD. All chemicals were of analytical grade.

### 3.2. Preparation of Galloyl Phytol, Galloyl Eicosanol, and Galloyl Phytanol

Galloyl phytol, galloyl eicosanol, and galloyl phytanol were synthesized according to the previously established method through a mild Steglich esterification [16]. Triisobutyryl gallic acid (3 mmol), phytol (phytanol or 1-eicosanol) (2 mmol), and DMAP (0.1 mmol) were dissolved in 12 mL toluene, followed by the addition of a solution of DCC (618.99 mg, 3 mmol) in 2 mL toluene. The mixture was stirred at an ambient temperature for 2 h. After that, the reaction mixture was added 14 mL 95% ethanol, followed by the addition of 18 mmol hydrazine hydrate (80%), and stirred further for 1 h. Then, the reaction mixture was acidified using hydrochloric acid (1N), extracted with ethyl acetate, and washed using water three times. The crude product was purified over a silica gel chromatography eluted with petroleum ether/ethyl acetate (1:1, *v*/*v*).

### 3.3. Structural Determination

The ^1^H and ^13^C NMR, FT-IR spectra, and mass spectra were determined to identify the structure of galloyl phytol, galloyl phytanol, and galloyl eicosanol. The ^1^H and ^13^C NMR of the prepared gallic acid esters were performed on a 400 MHz NMR spectrometer (Bruker Corporation, Fällanden, Zürich, Switzerland) at room temperature, respectively, employing DMSO-d6 (^1^H = 2.50 ppm, ^13^C = 39.52 ppm) as solvents. [21] FT-IR analysis was performed on an AVA TAR370 spectrophotometer (Thermo Nicolet Corporation, Madison, WI, USA) applying the attenuated total reflectance method with the spectral scanning scope of 400–4000 cm**^−^**^1^. The mass spectra were obtained on a Thermo Finnigan LCQ Deca XP Max system (Thermo Fisher Scientific, Waltham, MA, USA) employing a positive and negative ion electron spray ionization (ESI) mode with scan range of m/z 50–1500.

### 3.4. Thermal Gravimetry/Differential Thermal Analysis (TG/DTA) of Gallic Acid Esters

The TG/DTA analysis was employed to measure the thermodynamic property of gallic acid and gallic acid esters (galloyl phytol, galloyl phytanol, and galloyl eicosanol) on DTG-60A (Shimadzu Corporation, Kyoto, Japan). Around 2–5 mg of each sample was weighed and sealed in a hermetic aluminum pan, then heated from 10 to 600 °C at a rate of 10 °C/min with a hermetic empty aluminum pan as the control. Nitrogen was used as a transfer gas at a flow rate of 50 mL/min. The thermodynamic properties of samples were calculated from the thermal curve.

### 3.5. DPPH Scavenging Activity Assay

The DPPH scavenging activity was estimated according to the method described by Sánchez-Moreno et al. with slight modifications [22]. Briefly, 0.1 mL of methanol solution of different concentrations was added to 3.9 mL of 0.025 g/L DPPH methanol solution. Then, the absorbance of the mixture was measured at 515 nm on a UV-vis spectrophotometer (Model SP-756P, Shanghai Spectrum Corporation, Shanghai, China) after staying in dark for 30 min at an ambient temperature. The DPPH scavenging rate of the samples was analyzed during 0.5 h with an interval of 1 min at the concentration of 0.25 mM. The percentage of the scavenged DPPH (%DPPHSCA) was calculated according to the following equation. Methanol was applied as solvent and sample blank. BHA, BHT, and TBHQ were employed as positive control.
$$\left({\%\mathrm{DPPH}}_{\mathrm{SCA}}\right)=\left[1-\frac{A_{\mathrm{sample}}{-A}_{\mathrm{blank}}}{A_{\mathrm{control}}}\right]*100\%$$
where A_sample_ is the absorbance of the mixture of the antioxidation methanol solution (0.1 mL) and 0.025 g/L DPPH methanol solution (3.9 mL), A_blank_ is the absorbance of the mixture of antioxidation methanol solution (0.1 mL) and methanol (3.9 mL), A_control_ is the absorbance of the mixture of methanol (0.1 mL) and 0.025 g/L DPPH methanol solution (3.9 mL).

### 3.6. Inhibition of Lipid Oxidation in a Liposomal Model System

The antioxidant activity of gallic acid esters (galloyl phytol, galloyl phytanol, and galloyl eicosanol) on the lipid peroxidation was determined according to a previous study [23]. Liposomes were prepared using the thin-film hydration method. Lecithin (200 mg) was completely dissolved in chloroform (10 mL) and evaporated under vacuum at 25 °C. The lecithin film was preheated to 70 °C with a dropwise addition of 10 mL PBS (10 mM, pH 6.0) under magnetic agitation. Then, the mixture was stirred for 20 min to afford the fresh liposomes with a final lecithin concentration of 20 mg/mL.

To test the antioxidant activity, 0.5 mL sample at a fixed antioxidant concentration of 50 μM in a PBS buffer (0.01M, pH 6.0) was added to 1 mL liposome dispersion, followed by the addition of 0.5 mL AAPH (10 mM) solution. The mixture was incubated at 37 °C for 1 h. The PBS buffer was employed as blank. After the oxidation process, an aliquot (200 μL) of the mixture was withdrawn and reacted with 2-thiobarbituric acid (TBA) working liquid (300 μL) in the kit at 95 °C for 30 min. The reaction mixture was centrifuged at 13,000 rpm for 10 min after cooling down to an ambient temperature using ice. The absorbance values of supernatant (200 μL) at 532 and 600 nm were determined in a 96-well plate using a multimode microplate reader (Spark 10M, Tecan Group Ltd., Männedorf, Switzerland) and then the amount of thiobarbituric acid reactive substance was employed to estimate the antioxidant activity.

### 3.7. Inhibition of Lipid Oxidation in Oil

Azodiisobutyronitrile was applied as an oxidation reaction initiator to evaluate the antioxidant activity of gallic acid esters (galloyl phytol, galloyl phytanol, and galloyl eicosanol) in glycerol trioleate. The concentration of antioxidant and azodiisobutyronitrile in glycerol trioleate were 12.5 and 3 mM, respectively. Then, 5 μL of antioxidant ethanol solution was added to the test tube and streaming nitrogen was employed to remove the ethanol at room temperature. The mixture was incubated at 50 °C for 1 h and then the amount of produced thiobarbituric acid reactive substance was evaluated as above [24].

### 3.8. Statistical Analysis

All experiments were performed in triplicate and the result was expressed as mean ± standard deviation (SD). The least significant difference (LSD) in one-way analysis of variance (ANOVA) was employed to analyse the differences between the results applying the SPSS software (SPSS for Windows, SPSS Inc., Chicago, IL, USA). Differences with a *p* value of less than 0.05 were considered significant

## 4. Conclusions

In summary, a novel derivative of gallic acid with phytol was successfully prepared to employ the Steglich esterification. The integrated galloyl phytol maintained excellent radical scavenging activity and demonstrated outstanding antioxidant activity in dispersed and bulk oils. The TG/DTA, TEM, and molecular modeling investigations revealed that the branched aliphatic chain enforced the coiled molecular conformation and the unsaturated double bond in the phytol portion further fixed the coiled conformation, which contributed to diminished aggregation tendencies and enhanced antioxidant activities in dispersed and bulk oils. The remarkable antioxidant performance of galloyl phytol suggested intriguing antioxidant applications in the food industry, such as green synthetic nano silver for food packaging. The research and development of fat-soluble tea polyphenols as natural antioxidants not only make full use of tea resources and extend the tea industry chain but also meet the application needs of natural, non-toxic antioxidants.

## Figures and Tables

**Figure 1 molecules-27-07301-f001:**
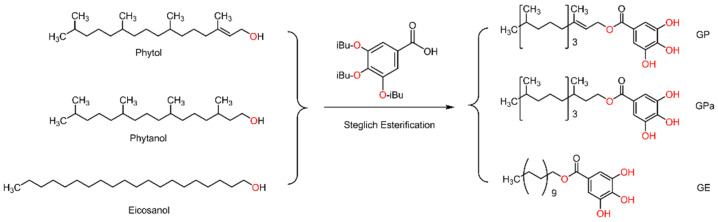
Synthetic route of galloyl phytol (GP), galloyl phytanol (GPa), and galloyl eicosanol (GE).

**Figure 2 molecules-27-07301-f002:**
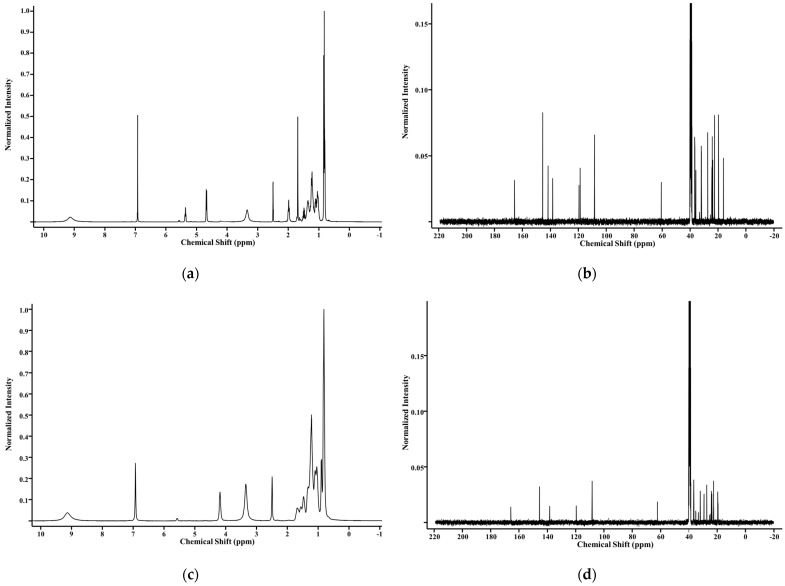
NMR spectrum of GP, GPa, and GE: (**a**) ^1^H-NMR spectrum of GP; (**b**) ^13^C-NMR spectrum of GP; (**c**) ^1^H-NMR spectrum of GPa; (**d**) ^13^C-NMR spectrum of GPa; (**e**) ^1^H-NMR spectrum of GE; (**f**) ^13^C-NMR spectrum of GE.

**Figure 3 molecules-27-07301-f003:**
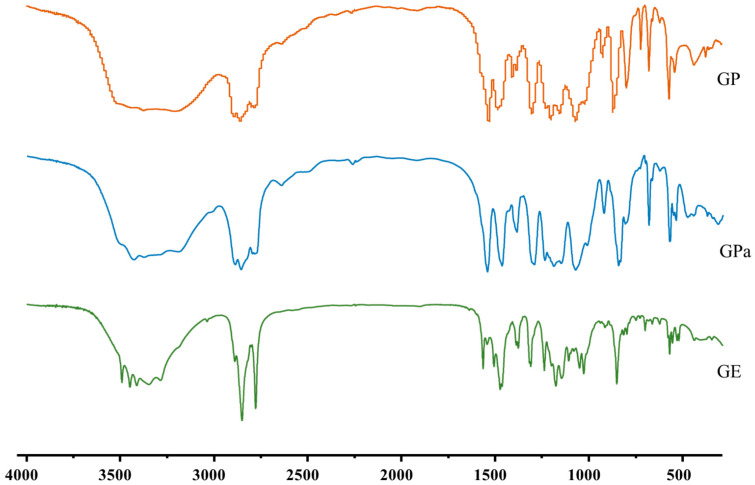
FT-IR spectrum of GP, GPa and GE.

**Figure 4 molecules-27-07301-f004:**
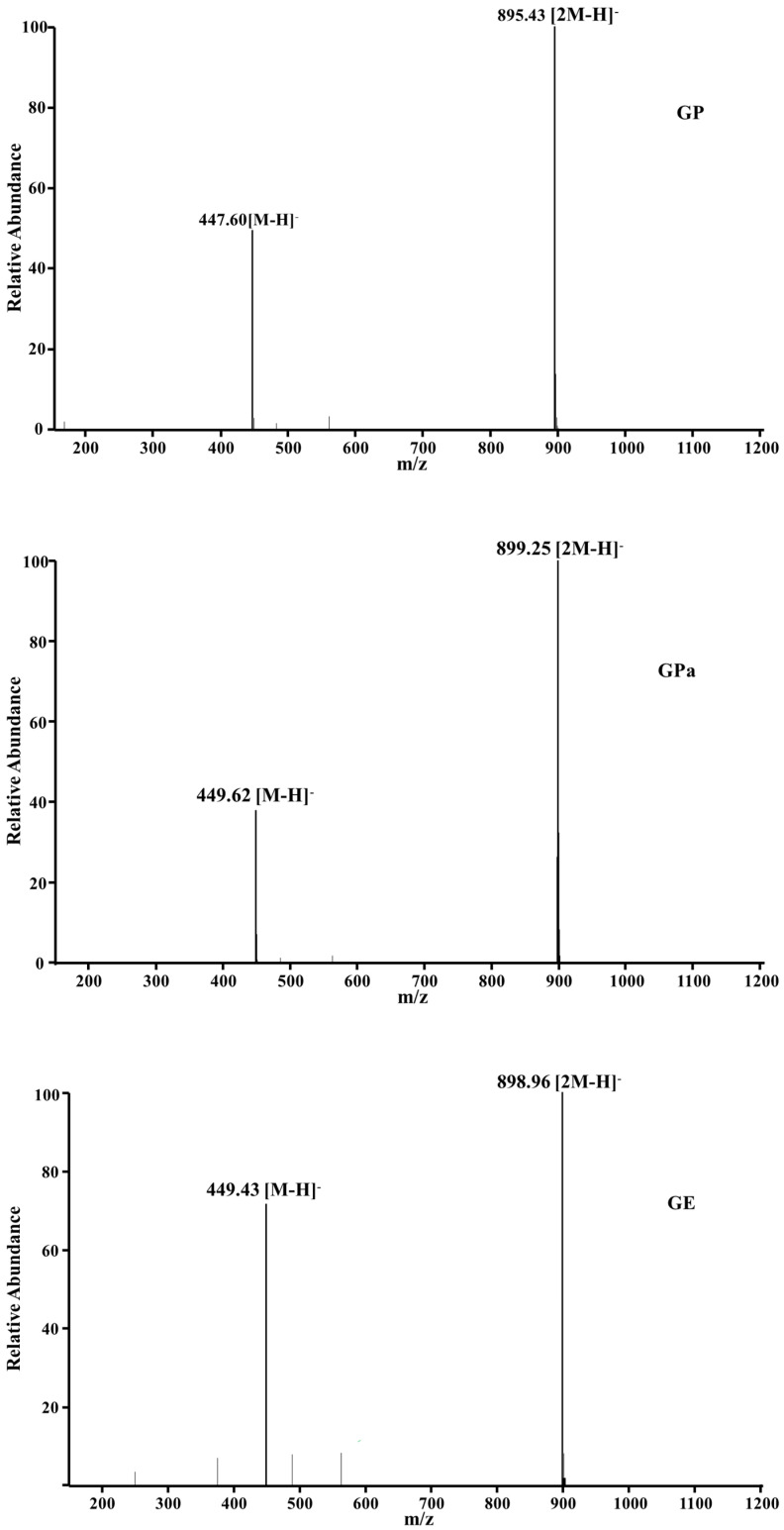
MS spectrum of GP, GPa, and GE.

**Figure 5 molecules-27-07301-f005:**
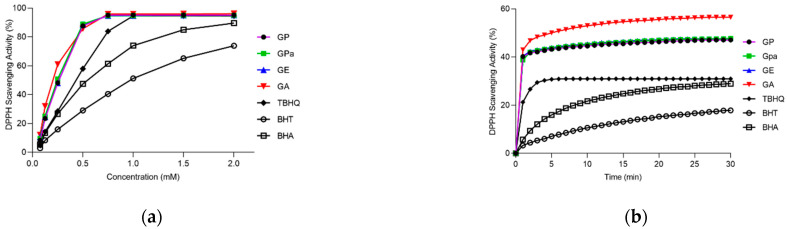
DPPH radical scavenging activity in different concentrations (**a**) and time (**b**).

**Figure 6 molecules-27-07301-f006:**
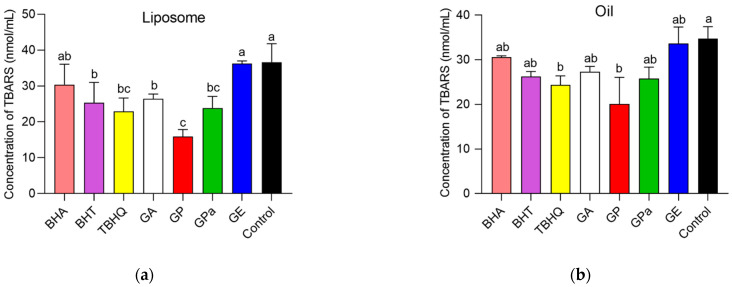
Concertation of thiobarbituric acid reactive substance (TBARS) in liposomal model (**a**) and glycerol trioleate bulk oil (**b**). Mean values with different letters are significantly different (*p* < 0.05). Different lowercase letters represent a significant difference (*p* < 0.05) between samples at same concentration.

**Figure 7 molecules-27-07301-f007:**
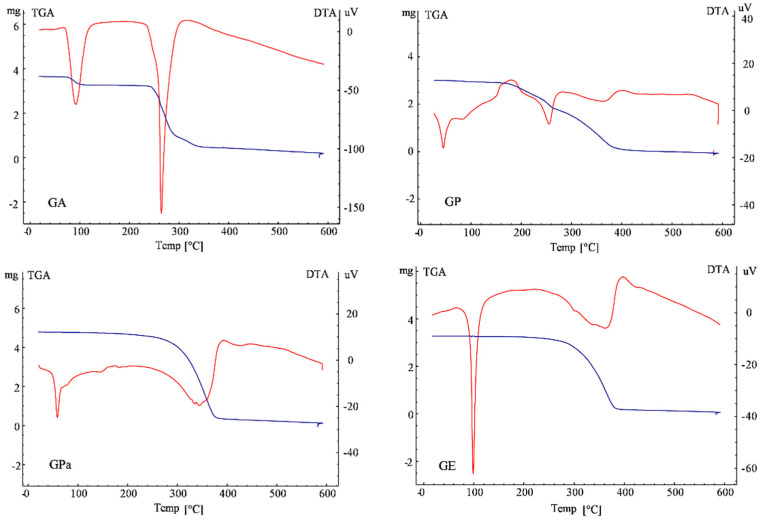
The TGA-DTA curve of gallic acid (GA), galloyl phytol (GP), galloyl phytanol (GPa) and galloyl eicosanol (GE). The blue line was the TGA curve, and the red line was the DTA curve.

**Figure 8 molecules-27-07301-f008:**
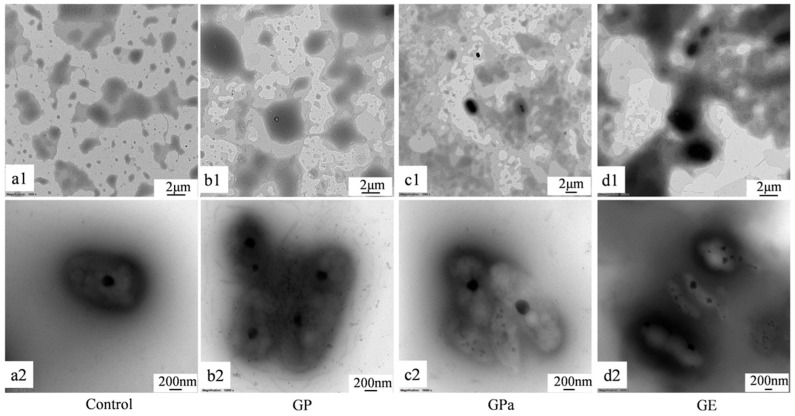
TEM images of the liposomes (**a1**,**a2**) with galloyl phytol (GP, (**b1**,**b2**)), galloyl phytanol (GPa, (**c1**,**c2**)), galloyl eicosanol (GE, (**d1**,**d2**)).

**Figure 9 molecules-27-07301-f009:**
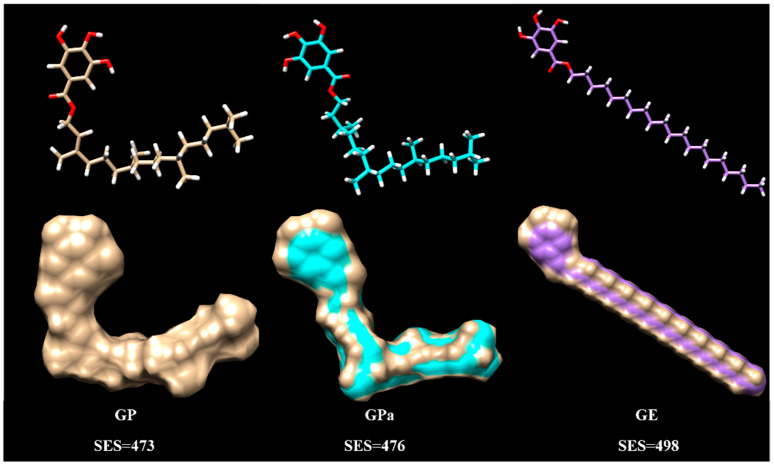
Molecular modeling of galloyl phytol (GP), galloyl phytanol (GPa), and galloyl eicosanol (GE).

## Data Availability

Not applicable.

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
