# Peer review of "Development of Galloyl Antioxidant for Dispersed and Bulk Oils through Incorporation of Branched Phytol Chain"

_molecules, 2022, doi:10.3390/molecules27217301_

Round 1

Reviewer 1 Report

The issue of the paper is interesting. The experiments are solidly prepared. The determination of the structure is solidly performed. Inhibition by phytol esters was confirmed. I propose the further continuation of the work.

Author Response

The issue of the paper is interesting. The experiments are solidly prepared. The determination of the structure is solidly performed. Inhibition by phytol esters was confirmed. I propose the further continuation of the work.

Response: Thank you for your kind comments.

Reviewer 2 Report

Title:

Development of Galloyl Antioxidant for Dispersed and Bulk 2 Oils through Incorporation of Branched Phytol Chain

Submitted Journal: Molecules

Abstract: it is too general in description please add some numerical data in the abstract part

Improve the quality of figure 2(b), 3 (b), and 4.

There is no 1H and 13C NMR, FT-IR spectra and mass spectra in the manuscript add  spectra along data to verify.

The extensive editing of english is required by native speaker.

This manuscript cannot be accepted in the present form.

Author Response

Point 1:

Abstract: it is too general in description please add some numerical data in the abstract part.

Reaponse 1: In the abstract part, we have added a numerical date. GP, GPa, and GE exhibited comparable DPPH scavenging activities, which were significantly higher than those of BHT, BHA, and TBHQ. The obtained EC50 values of GP, GPa and GE were 0.256, 0.262, 0.263 mM, respectively, which were much lower than that of TBHQ (0.431 mM), BHA (0.621 mM), BHT (0.956 mM).

Point 2: 

Improve the quality of figure 2(b), 3 (b), and 4.

Reaponse 2: The quality of Figure 2(b), 3 (b), and 4 have been improved.

Point 3: There are no 1H and 13C NMR, FT-IR spectra and mass spectra in the manuscript add spectra along with data to verify.

Response 3: The 1H and 13C NMR, FT-IR spectra, and mass spectra in the manuscript were added in Figure 4, and 5.

Point 4: The extensive editing of english is required by native speaker.

Response 4: The english writing has been improved.

Reviewer 3 Report

Dear Authors

The manuscript is design and presented very well. 

I wish to acceptance of manuscript after few corrections.

Authors need to write best value in the abstract section

improved quality of figure (3 and 4)

include future scope in conclusion section

Author Response

The manuscript is design and presented very well. I wish to acceptance of manuscript after few corrections.

Point 1: Authors need to write best value in the abstract section.

Response 1: We have added best value in the abstract section. The remarkable antioxidant performance of galloyl phytol suggested intriguing and non-toxic natural antioxidant applications in the food industry, such as effectively inhibition of the oxidation of oil and improvement of the quality and shelf life of the oil, which would contribute to the use of tea resources and extending the tea industry chain. 

Point 2: Improved quality of figure (3 and 4)

Response 2: The quality of figure 3 and 4 has been improved.

Point 3: include future scope in conclusion section

Response 3: Galloyl phytol has remarkable antioxidant performance, which can obviously inhibit oxidation rancidity and prolong the shelf life of the oil. Research and development of fat-soluble tea polyphenols as natural antioxidants, not only make full use of tea resources, and extend the tea industry chain, but also meet the application needs of natural, non-toxic antioxidants.